

# HtrA3: a promising prognostic biomarker and therapeutic target for head and neck squamous cell carcinoma

Yan Chen[1,2,3], Jianfeng Yang[1,2,3], Hangbin Jin[1,2,3], Weiwei Wen[4], Ying Xu[1,2,3], Xiaofeng Zhang[1,2,3] and Yu Wang[1,2,3]

[1] Department of Gastroenterology, Affiliated Hangzhou First People's Hospital, Zhejiang University School of Medicine, Hangzhou, China
[2] Key Laboratory of Integrated Traditional Chinese and Western Medicine for Biliary and Pancreatic Diseases of Zhejiang Province, Hangzhou, China
[3] Hangzhou Institute of Digestive Diseases, Hangzhou, China
[4] Department of Dermatology, Third People's Hospital of Hangzhou, Hangzhou, China

Corresponding authors
Xiaofeng Zhang,
zhangXiaofeng837@163.com
Yu Wang, wangyu@zcmu.edu.cn

## ABSTRACT

**Objective**. The dysregulation of the human high-temperature requirement A (HtrA) family of serine proteases is associated with many malignancies. However, there are few reports on HtrAs in head and neck squamous cell carcinoma (HNSCC). The aim of this study was to investigate the expression, prognostic value, and biological functions of HtrAs in HNSCC.

**Methods**. The RNA-sequencing data and clinical data of HNSCC were downloaded from The Cancer Genome Atlas (TCGA) database. The GSE30784 and GSE31056 datasets from the Gene Expression Omnibus (GEO) database were used for further verification. This study explored the differential expression of HtrAs and assessed their potential impact on the prognosis of HNSCC patients using a survival module. Correlations between clinical characteristics and HtrA expression levels were then explored using a Wilcoxon rank sum test. A Gene Ontology (GO), Kyoto Encyclopedia of Genes and Genomes (KEGG), and Gene Set Enrichment Analysis (GSEA) were performed using "clusterProfile" in the R software. A Pearson/Spearman correlation test was applied to analyze the relationship between HtrAs and immune infiltration level/checkpoint genes. Validation of HtrA expression levels were carried out by RT-PCR and western blot in human squamous carcinoma cell lines (Fadu and Cal-27) and human non-tumorigenic bronchial epithelium cells (BEAS-2B). Finally, through cell transfection, CCK-8, Ki-67 immunofluorescence, and flow cytometry assays, the effect of HtrA3 knockdown on the malignant biological behavior of HNSCC cells was explored.

**Results**. The gene expression levels of HtrAs were significantly upregulated and associated with patient age, TNM stage, clinical stage, and TP53 mutation status in the TCGA-HNSCC cohort. High expressions of HtrA1/3 were associated with shorter overall survival, shorter progress-free interval, and lower disease-specific survival in HNSCC. A nomogram for HtrAs was constructed and validated. HtrA-related genes were significantly enriched in the immune response and cell apoptosis pathway. In addition, the expression of HtrAs showed significant correlations with B cells, M cells, DC cell infiltration, and immune infiltration checkpoint (CD276, TNFRSF14). Validation of HtrA expression was carried out by RT-PCR and western blot. Results of

*in vitro* experiments indicated that HtrA3 gene knockdown inhibits the proliferation of FaDu and Cal-27 cells while concurrently promoting apoptosis.

**Conclusions**. HtrA3 shows significant potential as both a prognostic marker and a promising therapeutic target for HNSCC, highlighting its relevance and importance in future research and potential clinical applications.

## INTRODUCTION

Head and neck squamous cell carcinoma (HNSCC) is an aggressive and highly immunosuppressive malignancy, which originates from the mucosal epithelium of the oral cavity, pharynx, and larynx (*Ferris, 2015*; *Cramer et al., 2019*; *Johnson et al., 2020*). In 2020, there were 930,000 new cases of HNSCC and 460,000 HNSCC deaths globally, making HNSCC the sixth most common cancer worldwide (*Johnson et al., 2020*; *Sung et al., 2021*). Alcohol and smoking are the main risk factors for HNSCC (*Wyss et al., 2013*; *Chow, 2020*). Human papillomavirus 16 (HPV-16) infection is another risk factor for HNSCC progression, especially in the oropharyngeal subgroup (*Chaturvedi et al., 2011*; *Gillison et al., 2015*). Current treatments for HNSCC are radiotherapy (CRT), surgery combined with CRT, targeted therapy, and immunotherapy (*Hecht et al., 2021*). For patients with recurrent/metastatic (R/M) HNSCC, immune checkpoint inhibitors (ICIs), such as Pembrolizumab, are the standard first-line treatment (*Ferris et al., 2016*; *Burtness et al., 2019*). Unfortunately, only about 18% of HNSCC patients benefit from ICI therapy (*Ferris et al., 2016*; *Burtness et al., 2019*), showing the urgent need for predictive prognostic biomarkers for identifying effective therapeutic regimens.

The high-temperature requirement A (HtrA) family of highly-conserved serine proteases was originally identified in *E. coli* (*Lipinska et al., 1989*; *Strauch & Beckwith, 1988*). In this family, HtrA1, HtrA2, HtrA3 and HtrA4, are most closely related to humans (*Skorko-Glonek et al., 2013*). They share a modular structure and one or two PDZ structural domains involved in substrate recognition and bind with the His-Asp-Ser catalytic triad (*Zurawa-Janicka et al., 2017*). In addition, HtrA family members with variable N termini containing Sec export signal peptides, transmembrane structural domains, or secretion-related self-processing structural domains, are considered to be associated with cancer cell localization (*Zurawa-Janicka et al., 2017*). HtrA family members play important roles in cell physiology, cancer, neurodegenerative pathologies and immune diseases (*Zurawa-Janicka, Skorko-Glonek & Lipinska, 2010*). High expression of HtrA1 is associated with poor prognosis in patients with metastatic melanoma (*Sotiriou et al., 2006*), ovarian cancer (*Iacobuzio-Donahue et al., 2003*), breast cancer (*Chien et al., 2009*), prostate cancer, and lung carcinoma (*Moriya et al., 2015*). Similarly, the upregulation of HtrA2 is associated with malignant progression and poor prognosis of epithelial ovarian cancer (*Zurawa-Janicka et al., 2012*). Dysregulation of HtrA3 and HtrA4 is also associated

with many malignancies, such as pancreatic cancer, breast cancer, and hepatocellular carcinoma (*Xu et al., 2012*; *Hu et al., 2019*; *Pruefer et al., 2008*; *Chen et al., 2014*). However, the prognostic value and potential mechanisms of HtrAs in HNSCC are still unclear.

This study comprehensively explored HtrA expression levels between normal and HNSCC tissues, assessed their prognostic value, and evaluated the relationship between HtrA expression and clinical features, immune cell infiltration, and immune checkpoints. The biological functions of HtrAs in HNSCC were elucidated by Gene Ontology (GO), Kyoto Encyclopedia of Genes and Genomes (KEGG) and gene set enrichment analyses (GSEA). Finally, western blot and RT-PCR were applied to verify the expression levels of HtrAs in Cal-27 and FaDu cell lines. This study reveals the important role of HtrAs in HNSCC, and provides a theoretical basis for the study of prognostic markers in HNSCC. An overview of the research design and procedures is presented in Fig. 1.

## MATERIALS & METHODS

### Data acquisition and processing

The RNA-sequencing (RNA-seq) data and clinical data of HNSCC were downloaded from the TCGA database (*Hutter & Zenklusen, 2018*; *Colaprico et al., 2016*) (https://portal.gdc.cancer.gov/). The FPKM data were normalized using the transcripts per million (TPM) method and log2 transformed (*Shahriyari, 2019*). Then, the GSE30784 (*Chen et al., 2008*) and GSE31056 (*Reis et al., 2011*) datasets from the Gene Expression Omnibus (*Clough & Barrett, 2016*) (GEO, https://www.ncbi.nlm.nih.gov/geo) database were used to verify the differential expression levels of HtrAs between normal and HNSCC tissues. The gene expression profile of GEO was generated on the Affymetrix Human Genome U133 Plus 2.0 Array platform (*Chen et al., 2008*; *Reis et al., 2011*).

### Differential analysis and genomic alterations of HtrAs

The expression levels of HtrA family genes were compared between tumor and tumor-adjacent normal tissues with data from the TCGA (*Hutter & Zenklusen, 2018*). This dataset included grouped samples (tumor tissues $n = 502$, normal tissues $n = 44$) and paired samples ($n = 43$). A Wilcoxon rank sum test was used for the differential expression analysis of group samples, while a Wilcoxon signed rank test was used for paired samples (*Um et al., 2017*). Next, the mutation of HtrAs was studied by analyzing genomic alteration types, alteration frequencies, and protein changes in amino acids through the cBioPortal database (*Gao et al., 2013*) (http://cbioportal.org).

### Survival analysis and clinical characteristic analysis

The corresponding clinical prognosis information [overall survival (OS), disease-specific survival (DSS), and progress free interval (PFI)] was obtained from the TCGA dataset. Survival distributions were visualized using Kaplan–Meier curves, with samples dichotomized into high and low HtrA expression groups (*Győrffy, 2021*). The relationships between HtrA expression levels and clinical characteristics were analyzed using the Wilcoxon rank sum test (*Um et al., 2017*).

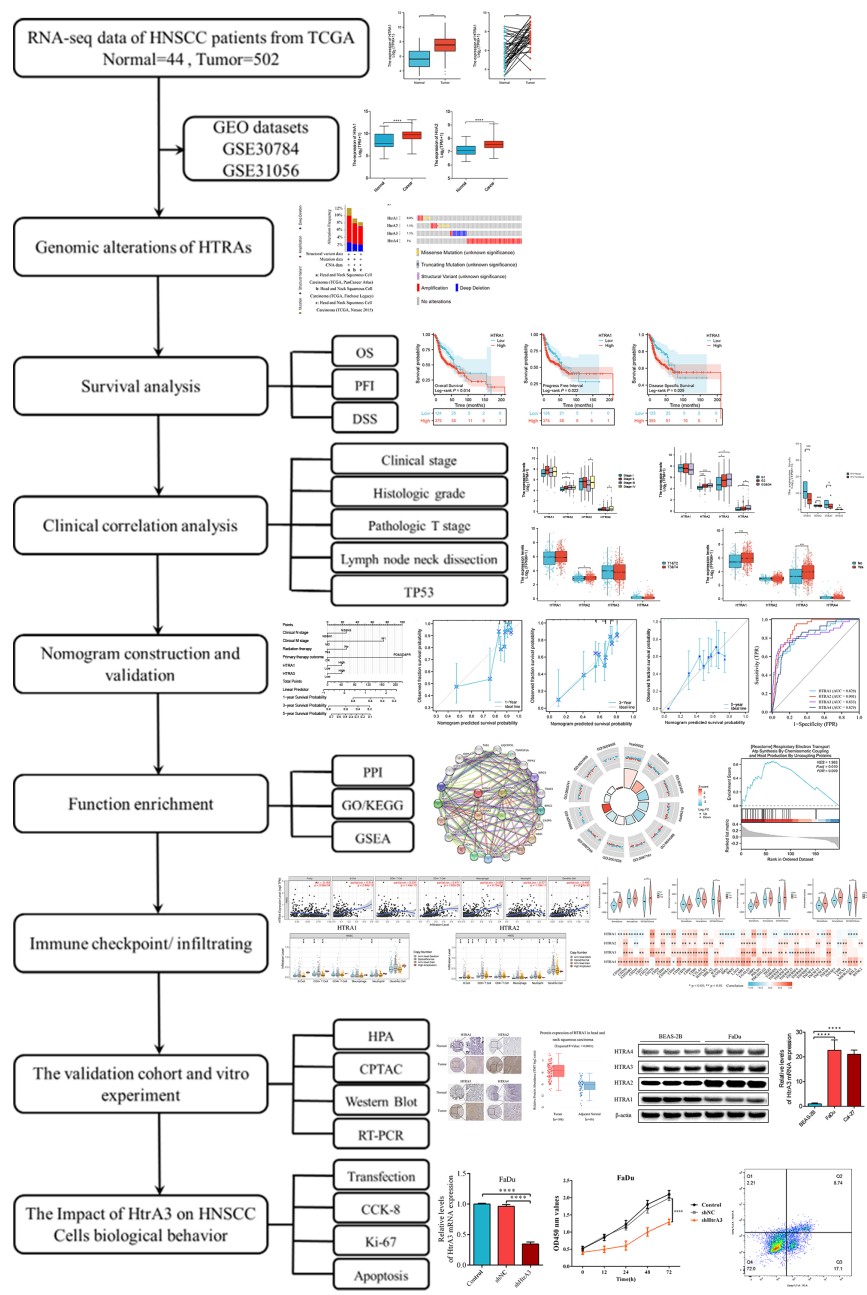

**Figure 1  Flowchart of this study.**

## Construction and validation of a nomogram

Univariate and multivariate analyses of HtrA expressions and clinical features (including age, TNM stage, clinical stage, radiation therapy, histologic grade, and primary therapy outcome) were performed to identify independent prognostic factors. Then the nomogram prediction model was constructed using the "rms package" in R (*Xu et al., 2021*). The calibrations were used to evaluate the prediction probability of the nomogram model (*Wang*

*et al., 2022*). To evaluate the prediction reliability of HtrAs, ROC curves were generated using the "Proc" package in R (*Robin et al., 2011*).

## Construction of PPI network and functional enrichment of HtrAs in HNSCC

The correlations between HtrA family genes were analyzed using a Pearson correlation test and visualized using "ggplot2" in R (*Liu et al., 2022*). We utilized the STRING (https://www.string-db.org/) online tool to construct a protein-protein interaction (PPI) network. The data was then imported into the Cytoscape software (http://www.cytoscape.org, version 3.9.1) for graphical optimization (*Shannon et al., 2003*). The GO, KEGG, and GSEA of HtrA-related genes were performed using the "clusterProfiler" package in R (*Yu et al., 2012*). Pathways enriched with adjusted $P < 0.05$, FDR $< 0.25$, and $|NES| > 1$ were considered to be significant.

## Correlation analysis between HtrA expression and immune checkpoint/infiltrating levels

Immune score and stromal score were calculated using "estimate" in R (*Yoshihara et al., 2013*). Correlations between HtrA expressions and six types of infiltrating immune cells in patients with HNSCC were analyzed using Tumor Immune Estimation Resource (TIMER) algorithms version 2.0 (*Li et al., 2017*) (https://cistrome.shinyapps.io/timer/). A Pearson correlation analysis was performed to evaluate the correlation of HtrA expression with immune checkpoint gene levels (*Newman et al., 2019*).

## Verification of differential expression of HtrAs in HNSCC

The UALCAN tool (*Chandrashekar et al., 2017*) (https://ualcan.path.uab.edu/) was used to verify protein differential expression between normal and HNSCC tissues in the CPTAC (Clinical Proteomic Tumor Analysis Consortium, https://proteomics.cancer.gov/programs/cptac) (*Ellis et al., 2013*). In addition, immunohistochemistry (IHC) images of HtrAs were obtained from HPA (*Navani, 2016*) (Human Protein Atlas, http://www.proteinatlas.org).

Human squamous carcinoma cell lines (Fadu and Cal-27) and human non-tumorigenic bronchial epithelium cells (BEAS-2B) (American Type Culture Collection, ATCC) were cultured in RPMI 1640 (Gibco BRL, Billings, MT, USA) and supplemented with 10% fetal bovine serum (FBS, Ex Cell) and antibiotics (100 U/Ml penicillin and 100 μg/Ml streptomycin) in a 5% CO2 incubator at 37 °C. Total RNA was extracted from cells using TRIzol (Thermo Fisher Scientific Inc., USA) and cDNA was synthesized using Prime Script™ RT Master Mixture (Thermo Fisher Scientific Inc., Waltham, MA, USA). RT-PCR was performed in triplicate with the SYBR-Green Master Mix (Thermo Fisher Scientific Inc.). Primers were purchased from Shanghai Sangon Biological Engineering Technology (Shanghai, China). The primer sequence is listed in Table S1. Relative expression levels were evaluated using the $2^{-\Delta\Delta Ct}$ method and GAPDH served as an internal control (*Livak & Schmittgen, 2001*).

Cells were lysed with protein extraction reagent (Beyotime, Beijing, China) supplemented with PMSF. The concentration of total protein was measured using the BCA Protein Assay Kit (Solarbio Life Science & Technology Company, Beijing, China) according to

the manufacturer's protocol. An equal amount of protein was separated in a 10% SDS polyacrylamide gel and then transferred to a PVDF membrane (Millipore, Burlington, MA, USA). After blocking with 5% skim milk for 2 h at room temperature, the membrane was incubated with primary antibodies against HtrA1 (55011-1-AP; Proteintech), HtrA2 (15775-1-AP; Proteintech), HtrA3 (bs-18100R; Bioss), HtrA4 (bs-18101R; Bioss), and $\beta$-actin (AM1021B, abcepta) at 4 °C overnight and then washed in TBST three times. The membranes were incubated with HRP-conjugated goat anti-rabbit IgG (1:1000; Beyotime) at room temperature for 2 h and then washed in TBST three times. Protein expression levels of the HtrAs were detected by ECL STAR luminous solution (Beyotime) using the Aplegen Gel Documentation System (OmegaLum G; GMI, Phoenix, AZ, USA).

## The impact of HtrA3 on Proliferation and Apoptosis in HNSCC

*Cell transfection.* Small hairpin RNA (shRNA) targeting HtrA3 and having a sequence of—CACACGGTTCCTCACAGAGTTTCAAGAGAACTCTGTGAGGAACCGTGTGTT—was constructed and inserted into the vector pGPU6/GFP/Neo-HtrA3-Homo. Two packaging plasmids (psPAX2 and pMD2.G), pGPU6/GFP/Neo-HtrA3-Homo (shHtrA3), and control vectors (shNC) were transfected into 293T cells for 48 h using Lipofectamine 3000 (Invitrogen, Waltham, MA, USA). Lentiviral particles were harvested and filtered for infection of FaDu and Cal-27 cells in combination with polybrene. Stably transfected cells were selected by neomycin. Transfection efficiency was verified by RT-PCR, and then a series of further experiments were carried out.

*CCK-8 assay.* The Cell Counting Kit-8 (CCK-8, Bimake, Houston, TX, USA) was used to assess the proliferative capacity of cells. Cells were inoculated into 96-well plates at 2,000 cells per well. Five replicated wells were set up for each group. Then, 10µl CCK8 solution was added to the wells and the samples were incubated for 1 h at 37 °C. The absorbance of the samples was measured for three consecutive days in a BioRad microplate reader (Bio-Rad, Hercules, CA, USA) at 450 nm.

*Ki-67 cell immunofluorescence.* Cells were fixed by one mL of 4% paraformaldehyde and 0.5% Triton X-100 was used for cell penetration at room temperature (RT, 25 °C) for 10 min. Cells were then washed with PBS and blocked with 3% BSA at room temperature for 1 h. The primary antibody (Anti-ki67 Rabbit antibody, 1:100) was added to each well and the cells were incubated overnight in a humidified cabinet at 4 °C. The secondary antibody (Cy3-conjugated Goat anti-Rabbit IgG, 1:50) was then added and the cells incubated at RT in the dark for 1 h. DAPI (4′,6-diamidino-2-phenylindole) was then added and the cells were incubated in the dark for 10 min. Cells were then observed and images were captured under a fluorescence microscope. Cells showing KI67-positive fluorescence were considered actively proliferating cells. The number of viable and proliferating cells was determined and recorded using ImageJ software (version v.1.8.0).

*Flow cytometry.* Cell apoptotic percentage was tested using a PE Annexin V Apoptosis Detection Kit (BD Pharmingen, Franklin Lakes, NJ, USA) following the manufacturer's instructions. Collected cells were briefly re-suspended in 100 µl 1× binding buffer, followed

by the addition of 5 µl PE Annexin V and 5 µl 7-AAD staining solution. The cells were then gently vortexed and incubated for 15 min at RT in the dark. A total of 400 µl of 1× binding buffer was then added to each tube and the apoptotic cell percentage was measured using flow cytometry (Beckman Coulter, USA).

## Statistical analysis

R software (version 4.2.2) and SPSS software (version 24.0) were used to perform all statistical analyses. Data was tested for homogeneity of variances and normality using a Levene's test and a Shapiro–Wilk normality test, respectively. If the data was normally distributed and the variances were equal, then a statistical analysis was performed with the Student's t test; if the data failed to meet the assumptions of equal variances, a Welch's $t$-test was used; if the data failed to meet the assumption of parametric tests, it was subjected to a Wilcoxon rank sum test. A log rank test was used for the survival analysis. Cox regression was used for the univariate and multivariable analyses, and a Pearson correlation analysis was used to analyze correlation. Band intensities on the western blot were quantified using ImageJ software (version v.1.8.0) and $\beta$-actin was used as the internal control to determine relative HtrA protein levels. A $P$-value of less than 0.05 indicated statistical significance.

## RESULTS

### The expression levels of HtrAs were significantly upregulated in HNSCC

Compared to adjacent normal tissues, HtrA1-4 mRNA expression levels were significantly higher in HNSCC tissues in both grouped samples (Fig. 2A) and paired samples (Fig. 2B) from the TCGA database. The GSE30784 and GSE31056 datasets from the GEO database were used for further verification. As shown in Fig. 2C, the mRNA expression levels of HtrA1-4 were also significantly upregulated in HNSCC tissues ($P < 0.05$). The cBioPortal online tool was used to analyze HtrA1-4 in patients with HNSCC for each gene alteration. The results showed that amplifications, mutations, and deep deletions were detected in three HNSCC subtypes (Fig. 2D). The mutation rate was 7% in HtrA4, 1.1% in both HtrA2 and HtrA3, and only 0.8% in HtrA1 (Fig. 2E). These results indicated that the HtrAs were relatively conserved.

### Prognostic value of HtrAs in HNSCC

Results of the survival analysis showed that high expressions of HtrA1/3 in HNSCC were associated with shorter OS, PFI, and DSS. However, HtrA2/4 expressions were not significantly associated with OS or PFI, and HtrA2 expression was not significantly associated with DSS (Figs. 3A–3C). In contrast, a low expression of HtrA4 was correlated with shorter DSS in HNSCC (Fig. 3C).

Next, the relationships between the expression levels of HtrA1-4 genes and clinical features in patients with HNSCC were explored. The expression level of HtrA2 in patients in clinical stage III was significantly higher than that of patients in stage I, and the expression levels of HtrA3/4 in patients in stage IV were significantly higher than those in stage III (Fig. 3D). As the histological grade increased, the expression of HtrA2 also gradually

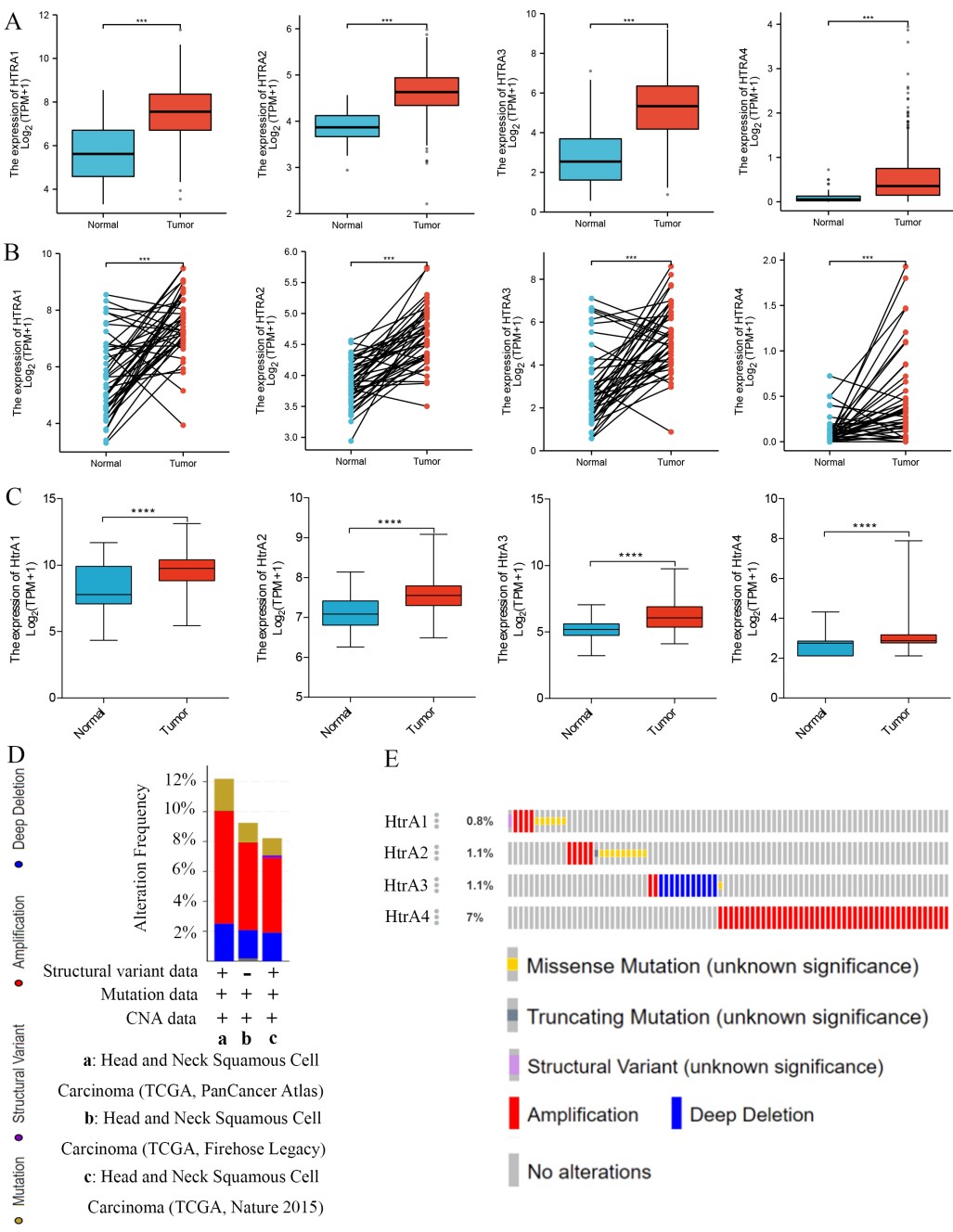

**Figure 2 The expression levels of HtrAs in HNSCC tissues and normal tissues.** (A) mRNA expression of HtrAs in grouped samples between HNSCC tissue ($n = 502$) and normal tissue ($n = 44$) from the TCGA database; (B) mRNA expression of HtrAs in paired samples between HNSCC tissue ($n = 43$) and normal tissue ($n = 43$) from the TCGA database; (C) Expression differences of HtaA1-4 in the GSE30784 and GSE31056 datasets; (D–E) genetic alterations in differentially expressed HtrA1-4 in HNSCC.

increased. The expression of HtrA3 in patients at G2/G3 & G4 was significantly higher than patients at G1. Similarly, compared with G1 and G2 patients, the expression of HtrA4 was significantly higher in patients at G3 & G4 (Fig. 3E). In patients with TP53 mutation,

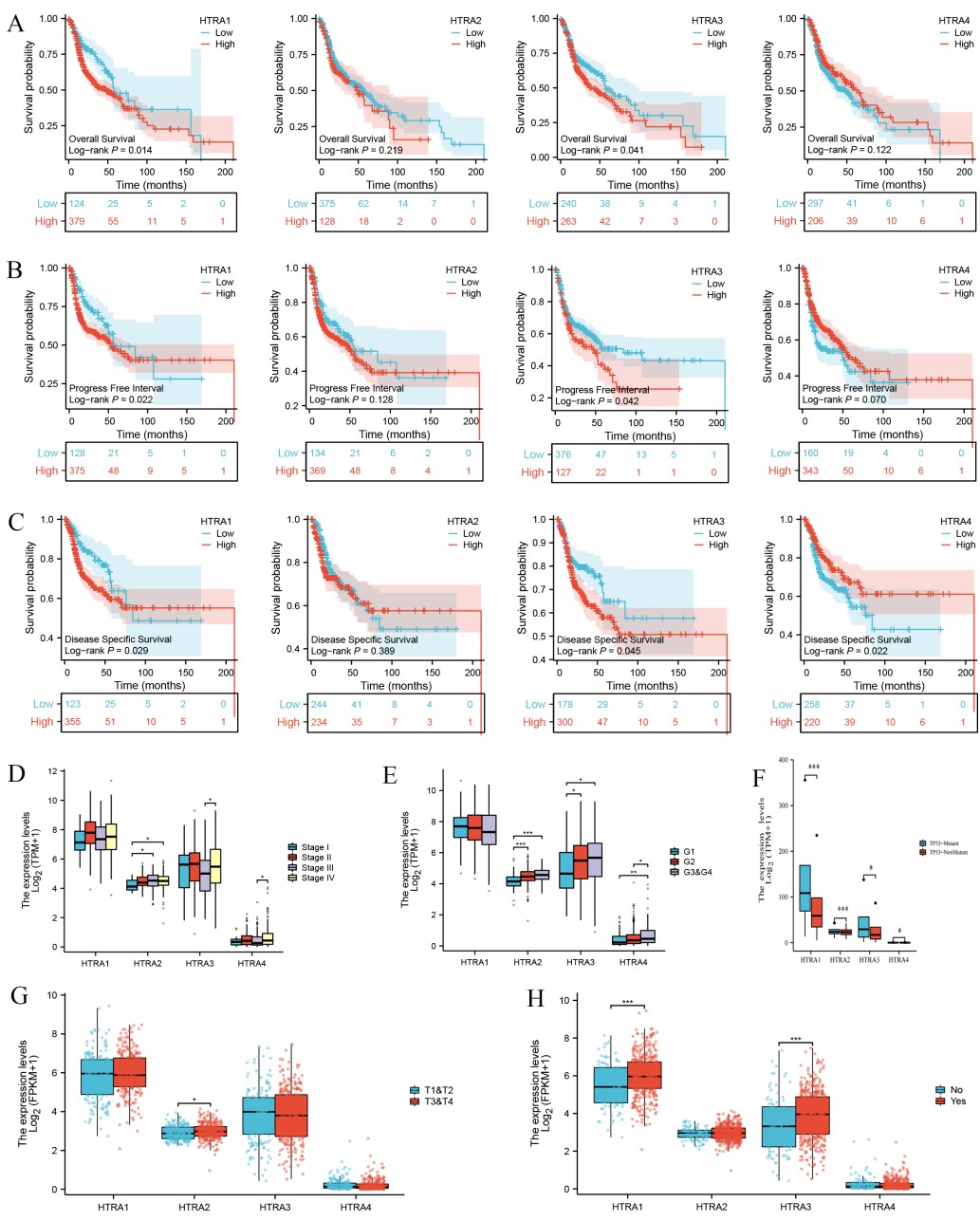

**Figure 3** **Correlation between HtrA1-4 expression and clinical characteristics of HNSCC and survival curves.** The association between high and low expression levels of HtrA1-4 and overall survival (A), progress free interval (B), and disease-specific survival (C). The relevance of HtrAs and clinical stage (D), histologic grade (E), TP53 mutation status (F), pathologic T stage (G), and lymph node neck dissection (H). * P < 0.05, ** P < 0.01, *** P < 0.001.

HtrA1-3 mRNA expressions were significantly higher than in those without TP53 mutation (Fig. 3F). In addition, compared to patients with T1 & T2, the expression of HtrA2 was significantly higher in patients with T3 & T4 (Fig. 3G). High expressions of HtrA1/3 were significantly associated with lymph node neck dissection (Fig. 3H).

## Construction and validation of a nomogram and ROC curve

Results of the univariate and multivariate analyses showed that HtrA1/3 were independent prognostic factors in HNSCC (Table 1). A nomogram was constructed to predict survival probability at 1, 3 and 5 years (Fig. 4A). The calibration plot for 1-, 3-, and 5-year OS demonstrated consistency between the predicted values by the nomogram and the actual values. The higher total points on the nomogram represented a worse prognosis. The calibration plots demonstrated that the nomograms were well-calibrated (Figs. 4B–4D). To further validate the performance of HtrAs, an ROC curve was plotted, and the area under the curve (AUC) of HtrA1-4 genes were 0.828, 0.901, 0.833, and 0.829, respectively (Fig. 4E), indicating that HtrAs could be potential diagnostic markers for HNSCC.

## Results of GO term, KEGG pathway enrichment, and GSEA

Results of the correlation analysis between HtrA family members showed that all HtrA genes were positively correlated with each other, except the HtrA1 gene and the HtrA4 gene (Fig. 5C). The top 20 genes closely related to HtrAs were extracted and constructed as a PPI network (Fig. 5A). In the GO biological processes (GO-BP) category, the HtrA-related genes were mainly enriched in the extrinsic apoptotic signaling pathway, apoptotic signaling pathway, and I-kappaB kinase/NF-kappaB signaling. In the GO cellular component (GO-CC) category, the HtrA-related genes were mainly associated with the cytochrome complex, mitochondrial outer membrane, and organelle outer membrane. In the GO molecular function (GO-MF) category, the most significant results were ubiquitin protein, ubiquitin-like protein ligase binding, and electron transfer activity (Fig. 5B, Table 2). According to the KEGG enrichment findings, the HtrA-related genes were significantly enriched in pathways of neurodegeneration—multiple diseases and apoptosis (Fig. 5B, Table 2). The GSEA revealed significant enrichment in respiratory electron transport atp synthesis by chemiosmotic coupling and heat production by uncoupling proteins (NES = 1.963, $P$.adj = 0.010), oxidative phosphorylation (NES = 2.026, $P$. adj = 0.010), respiratory electron transport (NES = 1.963, $P$.adj = 0.010), cardiac muscle contraction (NES = 1.922, $P$.adj = 0.025), and the citric acid TCA cycle, and respiratory electron transport (NES = 2.004, $P$.adj = 0.010; Figs. 5D–5H, Table 3).

## Expression levels of HtrAs were associated with multiple immune cells and immune checkpoints

To determine whether HtrAs were related to tumor immunity, this study further explored the correlation between HtrA expression levels and infiltration levels of six immune cell types as well as 46 immune checkpoint genes. Results indicated that the expression of HtrA1 was positively correlated with infiltration levels of CD4+ T cells, CD8+ T cells, dendritic cells (DC cells), and macrophage cells (M cells) in HNSCC. The expression of HtrA2 was positively associated with the abundance of CD4+ T cells, B cells, M cells, and DC cells. B cells, M cells, DC cells, CD4+ T cells, and neutrophils (NEUT) were the main immune cells affected by HtrA3 expression, while CD4+ T cells, B cells, CD8+ T cells, DC cells, NEUT cells, and M cells were more abundant in the high HtrA4 expression group (Figs. 6A–6B). The ESTIMATE algorithm was used to calculate immune score,

**Table 1 Univariate and multivariate Cox regression analyses and other clinicopathologic factors of OS in HNSCC.**

| Characteristics | Total (N) | Univariate analysis | | Multivariate analysis | |
|---|---|---|---|---|---|
| | | Hazard ratio (95% CI) | P value | Hazard ratio (95% CI) | P value |
| Age | 501 | | | | |
| <=60 | 245 | Reference | | | |
| >60 | 256 | 1.252 (0.956–1.639) | 0.102 | | |
| T stage | 486 | | | | |
| T1&T2 | 176 | Reference | | | |
| T3&T4 | 310 | 1.245 (0.932–1.661) | 0.137 | | |
| N stage | 479 | | | | |
| N0&N1 | 318 | Reference | | | |
| N2&N3 | 161 | 1.384 (1.040–1.842) | 0.026 | 1.498 (1.053–2.130) | 0.024 |
| M stage | 476 | | | | |
| M0 | 471 | Reference | | | |
| M1 | 5 | 4.745 (1.748–12.883) | 0.002 | 3.406 (0.469–24.756) | 0.226 |
| Clinical stage | 487 | | | | |
| Stage I &Stage II | 113 | Reference | | | |
| Stage III &Stage IV | 374 | 1.217 (0.878–1.688) | 0.238 | | |
| Radiation therapy | 440 | | | | |
| No | 153 | Reference | | | |
| Yes | 287 | 0.613 (0.452–0.831) | 0.002 | 0.669 (0.470–0.952) | 0.026 |
| Histologic grade | 482 | | | | |
| G1&G2 | 361 | Reference | | | |
| G4&G3 | 121 | 0.939 (0.688–1.282) | 0.692 | | |
| Primary therapy outcome | 417 | | | | |
| PR&PD&SD | 53 | Reference | | | |
| CR | 364 | 0.182 (0.124–0.268) | <0.001 | 0.196 (0.129–0.298) | <0.001 |
| HtrA1 | 501 | | | | |
| Low | 251 | Reference | | | |
| High | 250 | 1.309 (1.001–1.712) | 0.049 | 1.427 (0.986–2.064) | 0.060 |
| HtrA2 | 501 | | | | |
| Low | 250 | Reference | | | |
| High | 251 | 1.187 (0.909–1.550) | 0.207 | | |
| HtrA3 | 501 | | | | |
| Low | 251 | Reference | | | |
| High | 250 | 1.319 (1.008–1.726) | 0.044 | 1.312 (0.910–1.892) | 0.146 |
| HtrA4 | 501 | | | | |
| Low | 251 | Reference | | | |
| High | 250 | 0.894 (0.684–1.168) | 0.412 | | |

stromal score, and estimate score for each HNSCC patient. A high expression of HtrA1, HtrA3, and HtrA4 corresponded to a high immune score, stromal score, and estimate score (Fig. 6C). High HtrA2 expression was associated with a low stromal score and estimate score. In addition, HtrA1 was positively associated with CD276, CD44, and NRP1; HtrA3 was positively associated with CD200, CD276, CD80, CD86, HAVCR2, LAIR1, NRP1,

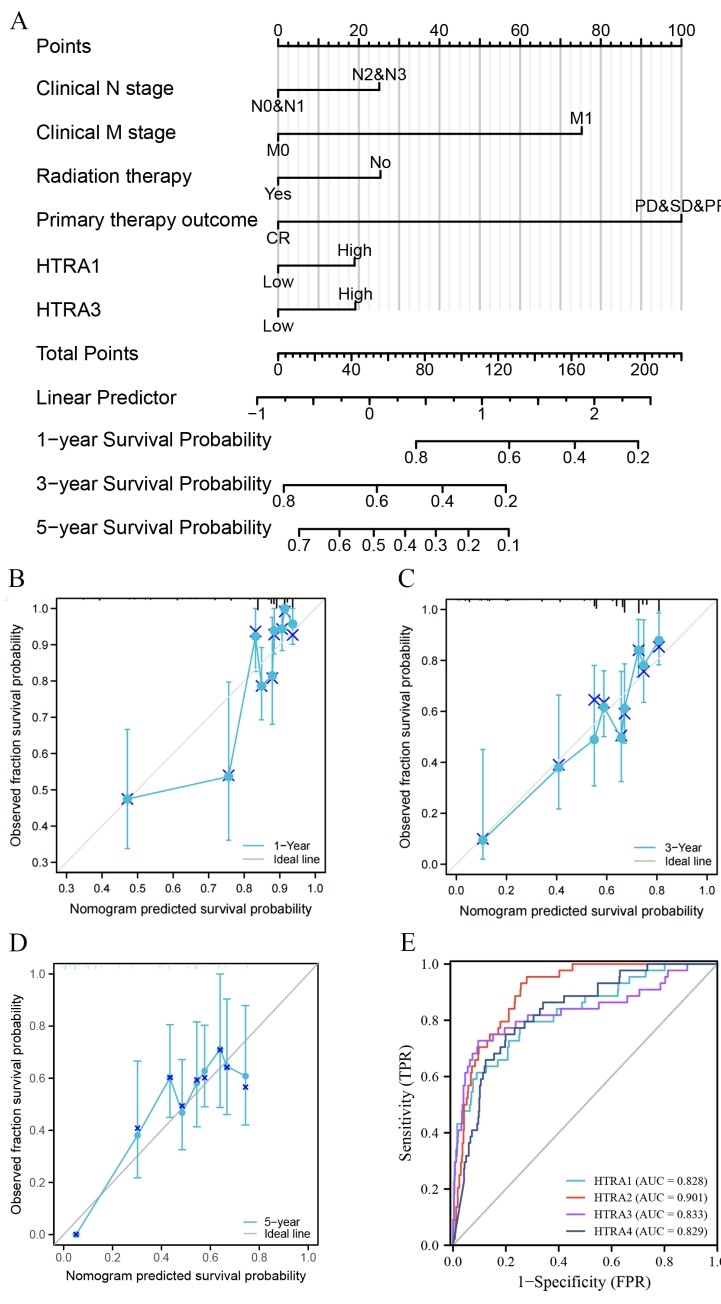

**Figure 4 Nomogram construction and validation.** (A) A nomogram to predict survival probability at 1, 3 and 5 years in patients with HNSCC; (B–D) the calibration curve of the nomogram; (E) receiver operating characteristic analysis (ROC) of HtrA1-4 in HNSCC patients.

TNFRSF4, TNFRSF9, and TNFSF4; and HtrA4 was positively associated with CD28, PDCD1, and TNFRSF14 (Fig. 6D).

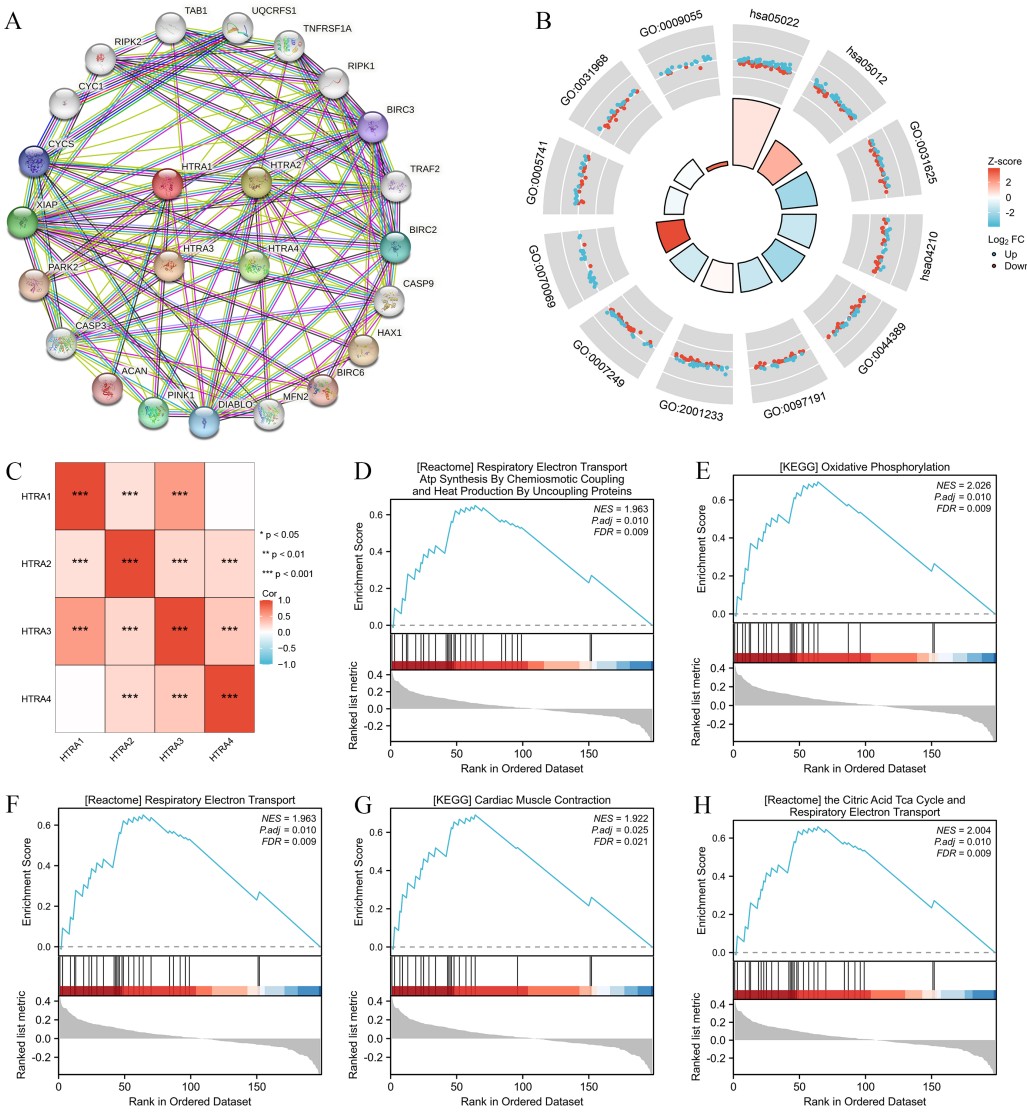

**Figure 5** **Genomic alterations and interactions of HtrA1-4 in HNSCC.** (A) PPI network analysis of HtrA family members and their 20 co-regulated hub genes conducted by STRING; (B) HtrA-related gene enrichment, pathway analysis, and functional profiles; (C) Pearson correlation analysis of HtrA1-4; (D–H) significant gene set enrichment analysis (GSEA) results of HtrAs including the hallmark pathways.

## Verification of HtrA expression in HNSCC

The HPA database showed that negative expressions of HtrA1-3 proteins were observed in normal tissues, while medium or high protein expressions were observed in HNSCC tissues. IHC revealed negative expression of HtrA4 in both normal and HNSCC tissues (Fig. 7A). Similarly, the protein expression levels of HtrA1 and HtrA3 were significantly higher in HNSCC tissues than in normal tissues in the CPTAC dataset (Figs. 7B–7C, $P < 0.0001$).

The expression levels of HtrA1-4 in HNSCC and non-tumor cells were also measured by RT-PCR and western blot verification. As shown in Fig. 7H, compared with normal cells,

**Table 2  HtrA-related gene enrichment, pathway analysis, and functional profiles.**

| Ontology | ID | Description | GeneRatio | BgRatio | *p* value | *p*.adjust | *z* score |
|---|---|---|---|---|---|---|---|
| BP | GO:0097191 | Extrinsic apoptotic signaling pathway | 38/198 | 221/18800 | 2.87e−35 | 1.05e−31 | −1.2977714 |
| BP | GO:2001233 | Regulation of apoptotic signaling pathway | 45/198 | 370/18800 | 6.61e−35 | 1.21e−31 | 0.1490712 |
| BP | GO:0007249 | I-kappaB kinase/NF-kappaB signaling | 41/198 | 288/18800 | 1.36e−34 | 1.65e−31 | −1.0932163 |
| CC | GO:0070069 | Cytochrome complex | 22/198 | 42/19594 | 1.64e−33 | 5.07e−31 | 3.8376129 |
| CC | GO:0005741 | Mitochondrial outer membrane | 32/198 | 205/19594 | 6.75e−29 | 1.04e−26 | −0.3535534 |
| CC | GO:0031968 | Organelle outer membrane | 33/198 | 232/19594 | 2.02e−28 | 2.08e−26 | −0.1740777 |
| MF | GO:0031625 | Ubiquitin protein ligase binding | 44/196 | 298/18410 | 1.09e−37 | 4.51e−35 | −2.1105794 |
| MF | GO:0044389 | Ubiquitin-like protein ligase binding | 44/196 | 317/18410 | 1.74e−36 | 3.61e−34 | −2.1105794 |
| MF | GO:0009055 | Electron transfer activity | 22/196 | 125/18410 | 7.82e−21 | 1.08e−18 | 2.9848100 |
| KEGG | hsa05022 | Pathways of neurodegeneration —multiple diseases | 79/170 | 476/8164 | 3.01e−53 | 6.23e−51 | 0.5625440 |
| KEGG | hsa05012 | Parkinson disease | 52/170 | 266/8164 | 2.27e−37 | 2.35e−35 | 1.6641006 |
| KEGG | hsa04210 | Apoptosis | 40/170 | 136/8164 | 4.09e−36 | 2.82e−34 | −1.2649111 |

**Table 3  Significant gene set enrichment analysis (GSEA) results of HtrAs, including hallmark pathways.**

| ID | SetSize | EnrichmentScore | NES | *p*value | *p*.adjust | *q*value |
|---|---|---|---|---|---|---|
| KEGG_OXIDATIVE _PHOSPHORYLATION | 24 | 0.6949955 | 2.026471 | 0.0001 | 0.0104 | 0.0086 |
| REACTOME_THE_CITRIC_ACID_TCA _CYCLE_AND_RESPIRATORY _ELECTRON_TRANSPORT | 29 | 0.6593472 | 2.004267 | 7.2e−05 | 0.0104 | 0.0086 |
| REACTOME_RESPIRATORY _ELECTRON_TRANSPORT | 28 | 0.6501578 | 1.962995 | 0.0002 | 0.0104 | 0.0086 |
| REACTOME_RESPIRATORY_ELECTRON _TRANSPORT_ATP_SYNTHESIS _BY_CHEMIOSMOTIC_COUPLING _AND_HEAT_PRODUCTION _BY_UNCOUPLING_PROTEINS | 28 | 0.6501578 | 1.962995 | 0.0002 | 0.0104 | 0.0086 |
| KEGG_CARDIAC_MUSCLE _CONTRACTION | 21 | 0.6910468 | 1.922093 | 0.0005 | 0.0251 | 0.0208 |
| WP_HOSTPATHOGEN_INTERACTION _OF_HUMAN_CORONAVIRUSES _APOPTOSIS | 13 | −0.7185709 | −1.972958 | 0.0016 | 0.0348 | 0.0288 |
| WP_FAS_LIGAND_PATHWAY_AND _STRESS_INDUCTION _OF_HEAT_SHOCK_PROTEINS | 17 | −0.6576734 | −1.968384 | 0.0016 | 0.0348 | 0.0288 |
| KEGG_RIG_I_LIKE_RECEPTOR _SIGNALING_PATHWAY | 15 | −0.6774612 | −1.952287 | 0.0016 | 0.0348 | 0.0288 |
| WP_NOVEL_INTRACELLULAR _COMPONENTS_OF_RIGILIKE _RECEPTOR_PATHWAY | 15 | −0.6774612 | −1.952287 | 0.0016 | 0.0348 | 0.0288 |
| KEGG_TOLL_LIKE_RECEPTOR _SIGNALING_PATHWAY | 20 | −0.6209131 | −1.945614 | 0.0019 | 0.0348 | 0.0288 |

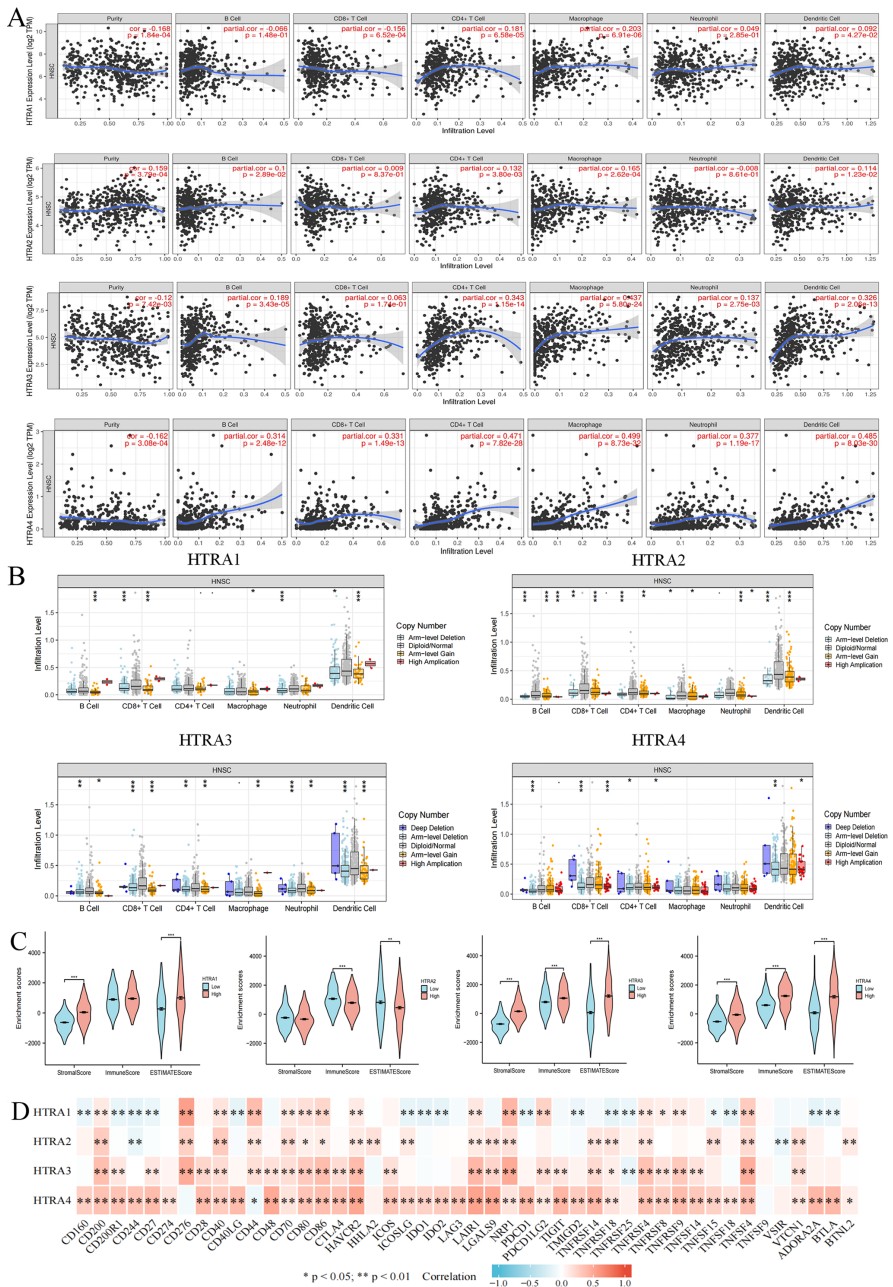

**Figure 6** **Associations between HtrA1-4 and the tumor immune microenvironment.** (A) TIMER database association of HtrA1-4 expression with immune infiltration level; (B) correlation of tumor-infiltrating levels in HNSCC and the alterations of different somatic copy numbers in HtrA1-4; (C) the correlation of HtrA1-4 expression level with immune, stromal and ESTIMATE score; (D) heatmap of the 46 common immune checkpoint genes and HtrA1-4 gene expression.

the mRNA expression levels of HtrA3 were significantly increased in FaDu and Cal-27 cells. As shown in Figs. 7D–7G, compared with normal cells, the protein expression levels of HtrA2/3/4 were significantly increased in FaDu and Cal-27 cells. These results all showed

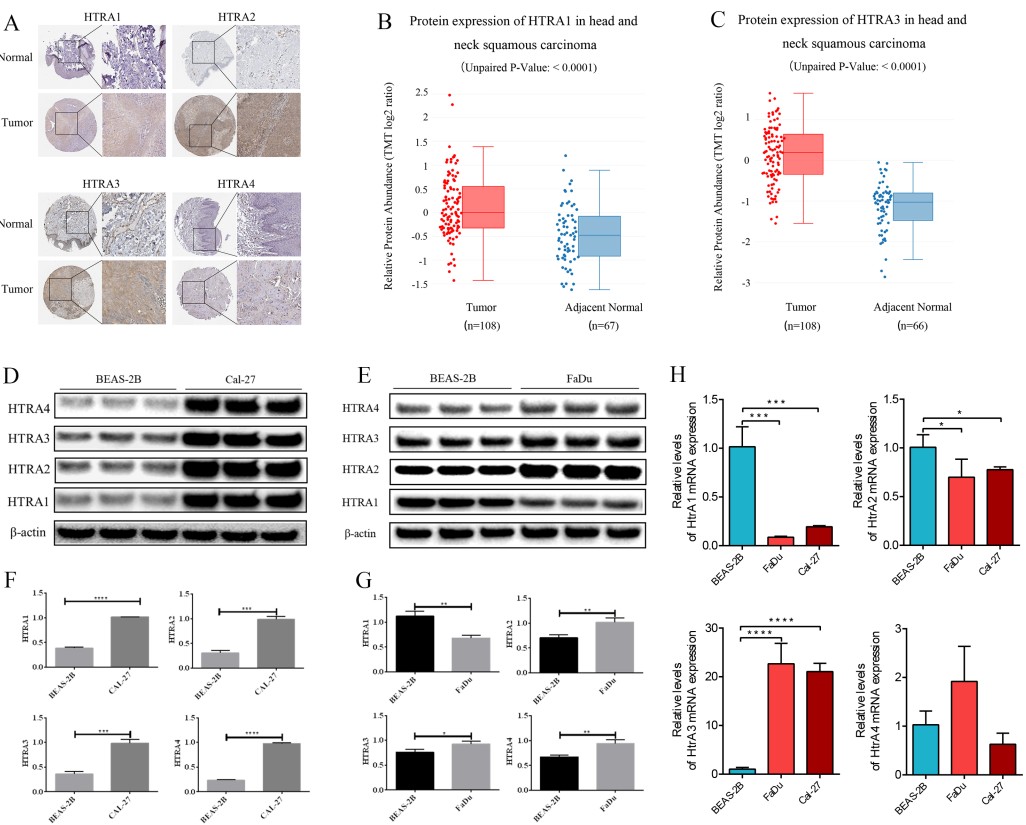

**Figure 7** **The validation cohort and *vitro* experiment.** (A) Representative immunohistochemistry images of HtrA1-4 in HNSCC and normal tissues (HPA database); (B–C) protein expression of HtrA1 and HtrA3 in HNSCC and normal tissues with datasets from the CPTAC database; expression of HtrA1-4 in BEAS-2B, Cal-27 (D) and Fadu (E) cells obtained by western blot. Values associated with test proteins were normalized to standard $\beta$-actin for the relative expression measure; (F–G) column graphs of western blot analysis; (H) HtrA1-4 expression in BEAS-2B, FaDu, and Cal-27 cells by RT-PCR. * $P < 0.05$, ** $P < 0.01$, *** $P < 0.001$, **** $P < 0.0001$.

that HtrA3 was upregulated in HNSCC at the gene level, while HtrA1-4 were upregulated in HNSCC at the protein level.

## The effect of HtrA3 knockdown on the biological behavior of HNSCC cells

After confirming the abnormal overexpression of HtrA3 in HNSCC, the influence of HtrA3 on the biological behavior of cells was explored. Efficient HtrA3 downregulation in FaDu and Cal-27 cells was verified by RT-qPCR analysis (Figs. 8A–8B). The CCK8 assay results showed that the optical density in the shNC group was higher than the optical density in the shHtrA3 group at 24 h, 48 h, and 72 h in FaDu and Cal-27 cell lines (Figs. 8C–8D). The results of the Ki-67 immunofluorescence assay indicated that among FaDu and Cal-27 cells, the percentage of Ki-67 fluorescence positive cells was higher in the shNC group compared to the percentage in the shHtrA3 group (Fig. 8E). The flow cytometry assay showed that the apoptotic cell percentage was markedly increased in FaDu and Cal-27 cells transfected

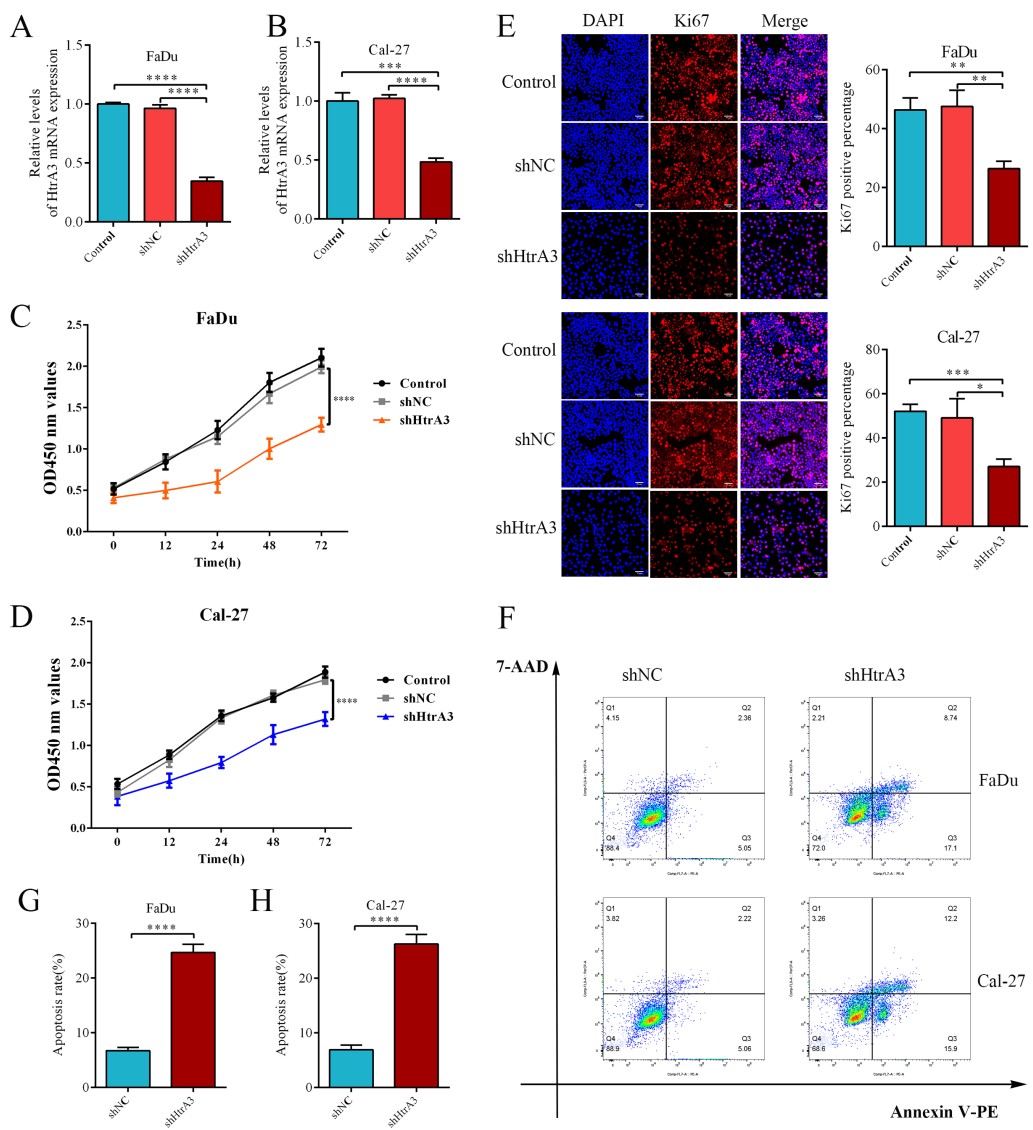

**Figure 8** **The effect of HtrA3 knockdown on the biological behavior of HNSCC cells.** (A–B) RT-PCR after cell transfection; (C–D) CCK8 experiment results of FaDu and Cal-27 cells after cell transfection. The OD values measured at 450 nm wavelength at 0 h, 12 h, 24 h, 48 h, and 72 h are displayed; (E) Ki-67 immunofluorescence staining of FaDu and Cal-27 cells after cell transfection. Ki67 red staining indicates cells in a state of proliferation, and DAPI blue nuclear staining represents the total number of living cells. The "Merge" image displays the combined view of both images. Relevant statistical graphs are also presented; (F–H) flow cytometry results and statistics related to apoptosis of FaDu and Cal-27 cells. $^*$ $P < 0.05$, $^{**}$ $P < 0.01$, $^{***}$ $P < 0.001$, $^{****}$ $P < 0.0001$.

with HtrA3 shRNA than in shNC-transfected FaDu and Cal-27 cells (Figs. 8F–8H). These results indicated that the knockdown of HtrA3 in FaDu and Cal-27 cells led to inhibited cell proliferation and the promotion of apoptosis.

## DISCUSSION

HNSCC is an aggressive malignancy with a high morbidity and mortality rate worldwide (*Ferris, 2015*; *Cramer et al., 2019*; *Johnson et al., 2020*; *Sung et al., 2021*). Although biomarkers for HNSCC have been identified, only a few of them have been used as targets for immunotherapy (*Ferris et al., 2016*; *Burtness et al., 2019*). Recent studies have shown that the dysregulation of HtrAs is associated with many malignancies, such as pancreatic adenocarcinoma, oral cancer, and breast cancer (*Sotiriou et al., 2006*; *Iacobuzio-Donahue et al., 2003*; *Chien et al., 2009*; *Moriya et al., 2015*). However, the prognostic value of HtrAs in patients with HNSCC has not yet been studied. This study identified the expression, prognostic value, and mechanism of HtrAs in HNSCC.

Bioinformatics analyses in this study showed that the mRNA expression levels of HtrAs in HNSCC were higher than in normal tissues. These findings are consistent with previous studies on the expression of HtrAs in gastric cancer, colon cancer, breast cancer, hepatocellular carcinoma, and malignant thyroid cancer (*Zurawa-Janicka et al., 2012*; *Xu et al., 2012*; *Hu et al., 2019*; *Pruefer et al., 2008*; *Chen et al., 2014*; *Wu et al., 2019*; *Ji et al., 2020*). Experimental verification results showed that HtrA3 expression was increased in HNSCC, consistent with other HNSCC databases (TCGA, GEO, HPA, CPTAC), and the RT-PCR and western blot results in this study. Our database results showed that HtrA1 expression was higher in HNSCC samples than in normal samples, but the RT-PCR results showed the opposite, and the western blot results showed that HtrA1 was highly expressed in Cal-27 cells but not in FaDu cells. The expression results of HtrA2 in the TCGA, GEO, and HPA databases and RT-PCR were inconsistent. TCGA and GEO databases and western blot results showed that HtrA4 expression was lower in normal samples than in HNSCC, but the HPA and CPTAC databases and RT-PCR results showed no statistically significant difference between the two. These conflicting results were likely due to patient, intra-patient, intra-tumor and intra-cell type heterogeneity (*Scott et al., 2016*).

Our study also found that HtrA expression was associated with patient age, TNM stage, clinical stage, radiation therapy, histologic grade, and TP53 mutation status in HNSCC. These results are consistent with those reported previously on HtrAs in gastric cancer (*Ji et al., 2020*). In addition, we observed that high expression levels of HtrAs were associated with poorer OS, PFI, and DSS in HNSCC. Since HtrA1 and HtrA3 were independent risk factors for HNSCC survival rates, a nomogram was constructed with comprehensive evaluation combining HtrA1 and HtrA3 with other important clinical features. To explain the potential molecular mechanisms by which HtrAs affect HNSCC prognosis, we performed GO and KEGG analyses and GSEA on HtrA-related genes. The results of the enrichment analysis showed that HtrAs were associated with the regulation of the apoptotic signalling pathway. The results from *in vitro* experiments further demonstrated that HtrA3 gene knockdown inhibits the proliferation of FaDu and Cal-27 cells while concurrently promoting apoptosis. These findings highlight the potential significance of HtrA3 as a key regulator in the growth and survival of HNSCC cells, suggesting its potential as a promising therapeutic target for further investigation and potential clinical applications.

HNSCC is among the most highly immune-infiltrated cancer types (*Mandal et al., 2016*). Tumor microenvironments with infiltrating immune cells can influence tumor development and progression (*Sun et al., 2019*). Infiltrating innate immunocytes in the tumor microenvironment may be associated with immune suppression (*Shao et al., 2020*; *Raju et al., 2022*; *Lisi et al., 2022*; *Li et al., 2022*; *Zuo, Zhao & Fan, 2022*). A recent study showed that HtrA3 plays a vital role in the progression of gastric cancer and that there was a positive correlation between HtrA3 expression and the abundances of innate immunocytes (natural killer cells, macrophages *etc.*) (*Ji et al., 2020*). Previous studies have also shown that IDO1 induction by HtrAs may contribute to the suppression of host immunity and facilitate carcinogenesis (*Clanchy et al., 2022*; *Wirthgen et al., 2018*; *Labadie, Bao & Luke, 2019*; *Boros & Vecsei, 2019*). HtrAs have been shown to be required for survival within macrophages (*Zhang et al., 2016*). In this study, we found that increased expression levels of HtrAs may promote the infiltration of CD4+ T cells, B cells, CD8+ T cells, NEUT cells, DC cells, and M cells. Together, these results indicate that immunosuppression induced by more innate immunocytes might result in a lower 1-, 3-, and 5-year survival rate in HNSCC patients with high expression levels of HtrAs. With the recent development of immune checkpoint inhibitors, biomarkers of immune cells can serve as prognostic markers (*Ladányi, 2015*). CD276 and NRP1 *etc.* showed a strong association with HtrA expression in HNSCC, supporting the important role of HtrAs in the immune contexture of HNSCC.

Our study also has limitations. First, our data primarily relied on information from public databases, which may not fully represent the diverse population of HNSCC patients. Second, while our study provided insights into molecular mechanisms through GO, KEGG, and GSEA, additional *in vivo* and *in vitro* experiments are necessary to fully elucidate these mechanisms. Based on our study results, we speculate that HtrAs might regulate the cell biological functions/immune infiltration process in HNSCC through the apoptotic signalling pathway, thereby affecting HNSCC prognosis. Our study provides new insights into HNSCC immunotherapy by investigating the prognostic role of HtrAs in HNSCC and identifies HtrA3 as a potential prognostic marker and promising therapeutic target for HNSCC.

## CONCLUSIONS

This study found that HtrA3 is highly expressed in HNSCC and is associated with a poor prognosis in HNSCC patients. HtrA3 is also correlated with immune infiltration. A HtrA3-related nomogram model was constructed and validated and HtrA3 was identified as a potential prognostic marker and promising therapeutic target for HNSCC. These results may help elucidate the role of HtrA3 in HNSCC based on clinical tumor samples.

## ACKNOWLEDGEMENTS

We are extremely grateful for reviewer comments that helped shape this manuscript.

## Funding

This work was supported by the Zhejiang Provincial Natural Science Foundation (No. LY21H270013 and No. LGF21H310004), the Medical and Health Technology Plan Project of Hangzhou (No. Z20230039), Zhejiang Medical and Health Science and Technology Plan (No. WKJ-ZJ-2136 and No. 2019RC068), and the Hangzhou Medical and Health Science and Technology Plan (No. 2016ZD01, No. OO20190610, and No. A20200174). The funders had no role in study design, data collection and analysis, decision to publish, or preparation of the manuscript.

## Grant Disclosures

The following grant information was disclosed by the authors:
Zhejiang Provincial Natural Science Foundation: LY21H270013, LGF21H310004.
Medical and Health Technology Plan Project of Hangzhou: Z20230039.
Zhejiang Medical and Health Science and Technology Plan: WKJ-ZJ-2136, 2019RC068.
Hangzhou Medical and Health Science and Technology Plan: 2016ZD01, OO20190610, A20200174.

## Competing Interests

The authors declare there are no competing interests.

## Author Contributions

- Yan Chen performed the experiments, analyzed the data, prepared figures and/or tables, authored or reviewed drafts of the article, and approved the final draft.
- Jianfeng Yang conceived and designed the experiments, authored or reviewed drafts of the article, and approved the final draft.
- Hangbin Jin analyzed the data, prepared figures and/or tables, and approved the final draft.
- Weiwei Wen analyzed the data, prepared figures and/or tables, and approved the final draft.
- Ying Xu performed the experiments, prepared figures and/or tables, and approved the final draft.
- Xiaofeng Zhang conceived and designed the experiments, authored or reviewed drafts of the article, and approved the final draft.
- Yu Wang conceived and designed the experiments, authored or reviewed drafts of the article, and approved the final draft.

## Data Availability

The data is available at NCBI GEO: GSE30784, GSE31056.
The raw measurements are available in the Supplemental Files.

## Supplemental Information

Supplemental information for this article can be found online at http://dx.doi.org/10.7717/peerj.16237#supplemental-information.

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
