# Peer review of "HtrA3: a promising prognostic biomarker and therapeutic target for head and neck squamous cell carcinoma"

_PeerJ, doi:10.7717/peerj.16237_

## Round 0.1 · original submission · Major Revisions

The reviewers have provided many key issues.

Reviewer 1 ·

Basic reporting

Some molecular characteristics of HtrAs family members, such as differential expression profiles, prognostic values, and potential mechanisms, have been explored in head and neck squamous cell carcinoma (HNSCC). Considering that there are no related literature reports, their results are very innovative and scientifically significant.

Experimental design

None.

Validity of the findings

None.

Additional comments

This article could be considered for publication after the following corrections.
1. Language editing services should be adopted to improve the quality of English.
2. Adding the references for all the publich datasets and bioinformatics algorithms.
3. Please provided the Specific Names for the GO/KEGG signaling in Figure 5B.
4. GO-BP/KEGG enrichment findings from Figure 5B indicated the functional roles of HtrAs in apoptosis. Thus, the effects of overexpression or targeted knockdown of HtrAs on cell proliferation and cell apoptosis should be explored in at least two HNSCC cells.
5. They said: “Enrichment analysis suggested that HtrAs genes may be involved in immune cell-related biological processes”. However, I couldn't find the enrichment findings for immune infiltration.
6. “Tumour microenvironment with infiltrating immune cells can influence tumour development and progression. Infiltrating innate immunocytes in the tumor microenvironment may be associated with immune suppression”. These discussions need to be supplemented with comprehensive references, like PMID: 35701781, PMID: 34826600, PMID: 35501802, PMID: 36115525, etc.

·

Basic reporting

The article must be written in English and must use clear, unambiguous, technically correct text. The article must conform to professional standards of courtesy and expression.

Experimental design

The submission should clearly define the research question, which must be relevant and meaningful. The knowledge gap being investigated should be identified, and statements should be made as to how the study contributes to filling that gap

Validity of the findings

The data on which the conclusions are based must be provided or made available in an acceptable discipline-specific repository. The data should be robust, statistically sound, and controlled.

Additional comments

I think the overall idea of the article is good, and I hope to add some tumor cell experiments for further verification.

·

Basic reporting

No comment

Experimental design

No comment

Validity of the findings

No comment

Additional comments

In this manuscript entitled “Expression, prognostic value, and mechanism of the HtrA family in head and neck squamous cell carcinoma”, Chen et al explored the expression and prognostic value of the HtrA family in HNSCC. It appears that the results of study are detailed and rich in content. However, some major concerns about this study still exist.
1. The title of this manuscript is not so convincing, as all the results were based on bioinformatic analysis of online databases and some in vitro experimental data, without in vivo experimental validation. The title seems exaggerate the results of this study, for the mechanism of the HtrA
family in head and neck squamous cell carcinoma was not demonstrated clearly.
2. The authors established a nomogram to predict survival probability at 1 and 3 years for HNSCC patients. However, the 5-year survival rate is a more important prognostic index for HNSCC. So, the survival probability at 5 years of the nomogram for HNSCC patients should be added.
3. I strongly suggest that your manuscript polished up by someone with expertise in technical English editing paying particular attention to English grammar and sentence structure, so that the manuscript will be more logical and clearer to the reader. For example,
1) Line 33-34: “We explored the differential 34 expression of HtrAs, assessed their potential impact on prognosis of HNSCC patients by survival 35 module. ” “and” is omitted.
2) Line 77:“Similarly, HtrA2 upregulated is associated with malignant progression and poor prognosis of epithelial ovarian cancer.” It should be “upregulation of HtrA2”
3) Line 82 “tissues, assessed their prognostic value, evaluated the relationship between HtrAs expression” . “and” is omitted.
4) Line 94 “The FPKM data was normalized using the”. It should be “were”, not “was”.
5) Line 95 “Then, GSE30784 and GSE31056 96 from the Gene Expression Omnibus” It should be “GSE30784 and GSE31056 96 datasets …….. were ……”
In addition, single and plural forms should be checked throughout the manuscript. The authors should check the WHOLE document for additional errors.

---

## Round 0.2 · accepted · Accept

Thank you very much for the careful revision by the authors. The current version is suitable for publication.

Reviewer 1 ·

Basic reporting

No comment

Experimental design

No comment

Validity of the findings

No comment

Additional comments

My questions have already been answered. I supported it to be accepted.

·

Basic reporting

No comment

Experimental design

No comment

Validity of the findings

No comment

Additional comments

No comment.

·

Basic reporting

no comment

Experimental design

no comment

Validity of the findings

no comment

Additional comments

no comment